# Natural Products Induce Different Anti-Tumor Immune Responses in Murine Models of 4T1 Mammary Carcinoma and B16-F10 Melanoma

**DOI:** 10.3390/ijms242316698

**Published:** 2023-11-24

**Authors:** Paola Lasso, Laura Rojas, Cindy Arévalo, Claudia Urueña, Natalia Murillo, Susana Fiorentino

**Affiliations:** Grupo de Inmunobiología y Biología Celular, Pontificia Universidad Javeriana, Bogotá 110231, Colombia; plasso@javeriana.edu.co (P.L.); rojasl.a@javeriana.edu.co (L.R.); cindy.arevalo@javeriana.edu.co (C.A.); curuena@javeriana.edu.co (C.U.); natalia.murillo@javeriana.edu.co (N.M.)

**Keywords:** breast cancer, melanoma, *Petiveria alliacea*, immunomodulation, antitumor, plant extracts

## Abstract

Natural products obtained from *Petiveria alliacea* (Anamu-SC) and *Caesalpinia spinosa* (P2Et) have been used for cancer treatment, but the mechanisms by which they exert their antitumor activity appear to be different. In the present work, we show that the Anamu-SC extract reduces tumor growth in the 4T1 murine mammary carcinoma model but not in the B16-F10 melanoma model, unlike the standardized P2Et extract. Both extracts decreased the levels of interleukin-10 (IL-10) in the B16-F10 model, but only P2Et increased the levels of tumor necrosis factor alpha (TNFα) and interferon gamma (IFNγ). Likewise, co-treatment of P2Et and doxorubicin (Dox) significantly reduced tumor size by 70% compared to the control group, but co-treatment of Anamu-SC with Dox had no additive effect. Analysis of intratumoral immune infiltrates showed that Anamu-SC decreased CD4+ T cell frequency more than P2Et but increased CD8+ T cell frequency more significantly. Both extracts reduced intratumoral monocytic myeloid-derived suppressor-like cell (M-MDSC-LC) migration, but the effect was lost when co-treated with doxorubicin. The use of P2Et alone or in co-treatment with Anamu-SC reduced the frequency of regulatory T cells and increased the CD8+/Treg ratio. In addition, Anamu-SC reduced glucose consumption in tumor cells, but this apparently has no effect on IFNγ- and TNFα-producing T cells, although it did reduce the frequency of IL-2-producing T cells. The efficacy of these herbal preparations is increasingly clear, as is the specificity conditioned by tumor heterogeneity as well as the different chemical complexity of each preparation. Although these results contribute to the understanding of specificity and its future benefits, they also underline the fact that the development of each of these standardized extracts called polymolecular drugs must follow a rigorous path to elucidate their biological activity.

## 1. Introduction

*Petiveria alliacea*, popularly known as Anamu, has been used for years in the treatment of cancer. It is widely distributed in nature, and its traditional use in the treatment of cancer occurs in various regions of the world, mainly on the American continent [1,2]. Anamu leaves have been extensively characterized phytochemically as well as for the diversity of additional uses of this plant [3,4,5]. Although dibenzyl trisulfide (DTS) has been considered the major active ingredient responsible for much of its antitumor activity, a drug containing this metabolite has not been developed due to its high toxicity and instability and its possible role as an inhibitor of intracellular cytochromes [6], suggesting that Anamu is the ideal candidate for the development of a polymolecular medicine.

We previously showed that an alcoholic extract of Anamu enriched with ethyl acetate exhibits antitumor activity in vitro against Mel-Ret cells of murine melanoma and A375 cells of human melanoma through the modification of the cell cytoskeleton, inducing arrest in G2 of the cell cycle, decreasing clonogenic capacity, and regulating proteins related to tumor metabolism [7]. These mechanisms of action are also exerted on 4T1 murine mammary carcinoma cells, where their ability to control the tumor is observed in BALB/c animals with orthotopic transplantation of 4T1 or murine TS/A mammary adenocarcinoma cells, possibly through the reduction of mitochondrial respiration and ATP production [8]. Furthermore, these extracts act preferentially on primary blasts from patients with acute myeloid leukemia (AML) [9], suggesting a cellular specificity that could be dependent on the intracellular microenvironment.

Our group previously obtained and standardized a gallotannin-rich fraction extracted from *Caesalpinia spinosa* and designated as P2Et. This extract has been characterized by its cytotoxic properties in several cancer cell lines, induction of immunogenic apoptosis through the expression of extracellular calreticulin, high-mobility group box 1 (HMGB1) protein, and ATP, modulation of the immune response, and in vivo antitumor effect in murine models of breast cancer and melanoma [8].

It has been proposed that the mechanism of action of botanical medications depends on the modulation of the tumor microenvironment, the immune response, and the control of tumor metabolism [10,11]. Although these alterations appear to be common in cancer, there are differences that affect the response to these types of new and old drugs. In the present work, we show that the Anamu-SC extract, which contains some of the compounds described as responsible for its in vivo activity, reduces tumor growth in the 4T1 murine mammary carcinoma model in BALB/c mice but does not reduce tumor in the B16-F10 melanoma model in C57BL/6 mice, in contrast to the standardized P2Et extract, which exhibits antitumor activity in both models as previously shown [8]. The anticancer efficacy of Anamu-SC does not appear to be related to immune response activation, so it also appears to be beneficial for P2Et and doxorubicin (Dox) therapy. 

## 2. Results

### 2.1. Chemical Composition of Anamu-SC

Ultra-High-Performance Liquid Chromatography (UHPLC) analysis of the extract showed a wide gamma of compounds with different intensities and polarities when identifying the peaks with the compounds previously reported for the plant. The compounds indicated in the chromatogram were identified by co-injection with commercial standards. The retention times observed for the marker compounds were 14.08 min for myricetin, 19.88 min for dibenzyl disulfide, and 21.25 min for dibenzyl trisulfide. The presence of myricetin in the extract was also confirmed through the analysis of the ultraviolet (UV) absorption spectra, where two considerable absorption bands were observed between the ranges 310–350 nm for band I and 250–280 nm for band II, characteristic of flavonols (Figure 1).

### 2.2. Comparative Anti-Tumor Activity

The Anamu-SC extract reduced 4T1 and B16-F10 cell viability in a dose-dependent manner with a half-maximal inhibitory concentration (IC_50_) of 125.05 ± 2.76 μg/mL and 85.8 ± 3.85 μg/mL, respectively (Appendix A), understood as the necessary concentration of the extract that causes a 50% reduction in cell viability or growth. P2Et extract also reduced cell viability with an IC_50_ of 34.1 ± 3.34 μg/mL (4T1) and 56.4 ± 1.50 μg/mL (B16-F10). Additionally, Anamu-SC with the IC_50_ and the fifth part of the IC_50_ (IC_50_/5) decreased the population doubling time (PDT), defined as the average duration of cell growth and division, in 4T1 and B16-F10 cells, suggesting that it may act as a cytostatic drug, perhaps by acting on the cell cytoskeleton and triggering cell cycle arrest, as previously shown [8]. Likewise, P2Et extract decreased PDT in both cell lines, as previously shown [12]. Intracellularly, Anamu-SC reduced glucose uptake in both cell lines, contrary to P2Et extract, which has a differential effect depending on the cell line, increasing glucose consumption on 4T1 without significantly changing it on B16-F10 (Figure 2C,D). Regarding antioxidant activity, Anamu-SC did not induce changes in 4T1 cells, unlike in the B16-F10 cell line, where it had pro-oxidant activity (Figure 2E,F). In turn, in both lines, the antioxidant activity of the P2Et extract was confirmed, as previously observed [12]. This result reflects the complexity of the biological activity of these extracts and their relationship with the cellular microenvironment, clearly showing that each polymolecular mix must be studied as an independent drug.

### 2.3. Anamu-SC Shows Specific Activity against 4T1 Murine Mammary Carcinoma Cells

Subsequently, we evaluated the in vivo activity of the Anamu-SC extract in two murine tumor models (Figure 3A), finding that the Anamu-SC extract did not reduce tumor growth in the B16-F10 melanoma model (Figure 3B,C), while it did in the 4T1 murine mammary carcinoma model (Figure 3D,E). The Anamu-SC extract decreased tumor growth by 55% compared to the PBS control, as did the P2Et-positive control (Figure 3D). As previously shown, P2Et extract showed unequivocal anticancer activity in both models [7,8].

The lack of antitumor activity of Anamu-SC extract in the B16-F10 murine model seems to be linked to a lack of modulation of the intratumorally immune response, defined as a low infiltration of CD45+ (Figure 3F) and CD8+ (Figure 3G) cells, a reduced frequency of CD4+ CD44+ and CD8+ CD44+ T cells (Figure 3H), as well as a relative increase in the frequency of monocytic myeloid-derived suppressor-like cells (M-MDSC-LC) and polymorphonuclear myeloid-derived suppressor-like cells (PMN-MDSC-LC) (Figure 3I), unlike those observed with P2Et extract. In contrast, in the 4T1 model, both Anamu-SC and P2Et increased the CD45+ intratumoral infiltrate (Figure 3J) as well as the frequency of CD8+ T cells, but interestingly, Anamu-SC reduced the frequency of CD4+ T cells (Figure 3K). Unlike the P2Et extract, treatment with Anamu-SC does not lead to the activation of CD8+ T cells recognized to express CD44 (Figure 3L), nor does it significantly reduce the frequency of PMN-MDSCs (Figure 3M). Taken together, these results show that although the Anamu-SC extract has antitumor activity in the 4T1 model, it is mainly immunosuppressive, which may further affect tumor growth control in the B16-F10 melanoma murine model. When evaluating plasma cytokines in the 4T1 model, we found that none of the extracts substantially modified the cytokine concentration. In contrast, the two extracts decreased IL-10 levels in the B16-F10 murine melanoma model, but only P2Et treatment increased TNFα and IFNγ levels, indicating increased activation of the effector immune response (Appendix A).

### 2.4. Intratumoral Immune Response Regulation in the Presence of Extracts

Chemotherapy activity, in particular the activity of anthracyclines, can be inhibited by natural compounds. Therefore, we evaluated the effect of the extracts individually and in combination with Dox in both murine models (Figure 4A). We observed that individual treatment with Anamu-SC or P2Et reduced tumors by 48% and 64%, respectively, compared to the control (Figure 4B,C). Likewise, co-treatment of P2Et and Dox significantly reduced tumor size by 70% compared to the control group. However, with the co-treatment of Anamu-SC and Dox, no differences were found compared to each treatment administered individually (Figure 4B,C). When evaluating the type of intratumorally immune infiltrate, we confirmed that Anamu-SC decreases the frequency of CD4+ T cells more than P2Et but increases the frequency of CD8+ T cells more significantly (Figure 4D). When co-treatment was evaluated, P2Et tended to reduce the frequency of intratumoral CD4+ T cells in the presence of Dox, but there was no significant change in the frequency of CD8+ T cells with either extract (Figure 4D) or Tregs present in response to Dox treatment (Figure 4E).

Interestingly, Anamu-SC alone elicited more DC infiltration than any other treatment (Figure 4F), but unlike P2Et, these DC are not CD8α+, which are not involved in cross-priming (Figure 4G). In fact, neither Anamu-SC nor Dox alone or in co-treatment with any of the extracts induce CD8α+ DCs. When analyzing the effect of the extracts on the reduction of populations known as immunosuppressive, the reduction of M-MDSC-LC with both P2Et and Anamu-SC is more notable, with a slight trend for PMN-MDSC-LC. This effect was lost when co-treatment with Dox was performed, although not for PMN-MDSC (Figure 4H), suggesting that the effect of Anamu-SC may involve a reduction of this population with an increase in macrophages that could also be immunosuppressive (Figure 4I), and that would explain why the antitumor effect of Dox decreases in the presence of Anamu-SC.

### 2.5. Anamu-SC Extract Subtly Potentiates the Antitumor Activity of P2Et Extract

In order to further explore possible interactions, not only with chemotherapeutics but also with botanical extracts, we evaluated the effect of co-treatment with P2Et and Anamu-SC in the 4T1 murine mammary carcinoma model (Figure 5A). In addition to the previously reported effect of the P2Et extract, we observed that co-treatment with the two extracts substantially decreased tumor size by 67% compared to the control group (Figure 5B,C).

Furthermore, P2Et, either alone or in combination with Anamu-SC, decreased the occurrence of macrometastasis compared to the control group (Figure 5D). When evaluating the intratumoral immune response, the use of the combined extracts significantly reduced the frequency of CD4+ T cells and increased CD8+ T cells to a greater extent (Figure 5E). No differences were found in the frequency of activated CD4+ T cells assessed by CD44 expression, but differences were found in CD8+ T cells (Figure 5F). However, a lower frequency of Tregs was found when P2Et was used alone or in combination with Anamu-SC compared to the control group (Figure 5G). Likewise, the CD8+/Treg ratio was higher when P2Et was used alone or in co-treatment with Anamu-SC (Figure 5H). As previously observed, P2Et increased DC migration even in the presence of Anamu-SC, but, interestingly, we again found that Anamu-SC does not favor CD8α expression on these cells, which may mean that this DC population does not migrate to the tumor when Anamu-SC is present or that this extract inhibits CD8α expression on DCs (Figure 5I). We also found that co-treatment reduced the presence of PMN-MDSC PD-L1+ but did not generate changes in the M-MDSC PD-L1 population (Figure 5J).

When evaluating the effect of the extracts on intracellular cytokine production, we observed that co-treatment with P2Et and Anamu-SC reduced the frequency of CD4+ T cells producing IFNγ that was induced by treatment with P2Et extract alone but did not alter the frequency of CD4+ T cells producing TNFα and IL-2 (Figure 6A). In contrast, in CD8+ T cells, both P2Et and co-treatment increased the frequency of CD8+ T cells producing IFNγ or TNFα, which may be related to the presence of intratumoral CD8+ T cells induced by the two extracts, but interestingly reduced the frequency of CD8+ T cells producing IL-2. Neither extract induced changes in the frequency of CD4+ or CD8+ T cells producing granzyme or perforin (Figure 6B).

The evaluation of T cell response, according to previous reports, should be based on its polyfunctional activity, known as the ability to produce two or more cytokines simultaneously [13]. To extend the assessment of functional T cell activity in each group of mice, we measured the simultaneous production of IFNγ, TNFα, IL-2, perforin, and granzyme B after stimulating the cells with P/I. In summary, co-treatment with P2Et and Anamu-SC induced a higher frequency of multifunctional CD4+ and CD8+ T cells compared to the control group, like that observed with mice treated with P2Et only (Figure 6C,D). However, no differences were found between the treatment with P2Et and the co-treatment with Anamu-SC, indicating that, at least in the improvement of the quality of the response, there is no synergistic behavior between the two extracts.

## 3. Discussion

In this work, we show that the Anamu-SC extract has preferential activity on the 4T1 mammary carcinoma model but not on the B16-F10 melanoma model, unlike the standardized P2Et extract (Figure 3). These results are interesting because they confirm that although activity was observed on tumor cells in vitro (Figure 2), this was not reflected in the in vivo model, suggesting again that the search for antitumor agents must consider not only activity on the tumor itself but also on the tumor environment. Differences in activity depending on the tumor model have been previously reported. For example, results with *Piper nigrum* L. show that although it shows positive effects in prostate and breast cancer, the lignans present in the methanolic extract and piperine show opposite effects in melanoma. That is, a significant stimulatory effect on melanogenesis in murine B16 melanoma cells [14], but an antitumor and apoptosis-inducing effect on human melanoma cells [15].

The intrinsic differences of each tumor have been evidenced in previous works. Ultrasound-guided pFUS treatment in 4T1 and B16-F10 tumors showed that after 3 days of treatment, a pro-inflammatory microenvironment was induced in both tumors, but less so in B16-F10. Treatment induces Ki-67 expression in B16-F10 but not in 4T1, even though pro-apoptotic caspase 3 activation is induced in both tumors. After 12 h of treatment, B16-F10 tumors become more proliferative and malignant due to activation of the epithelial mesenchymal transition process (EMT) and KRAS, unlike 4T1 tumors, where these pathways are not enriched in response to treatment [16]. The increase in EMT increases resistance to therapy, and in this case, the control of metastases by the immune system plays a very important role [17,18].

It is possible that a healthy and activated immune system is necessary for the control of melanoma rather than murine breast cancer. Since Anamu-SC extract does not stimulate the immune system per se, this could explain its lack of activity in the melanoma model. In the same direction, it has been recently shown that *Tillandsia usneoides* (*T. usneoides*) extract shows efficacy in the 4T1 mammary carcinoma model but not in the B16-F10 melanoma [12]. In both cases, Anamu-SC and *T. usneoides* extracts do not promote intratumoral migration of CD45 cells, and the presence of activated T cells is reduced. Differences in the type of cell death induced by each extract (P2Et, Anamu-SC, and *T. usneoides*) condition the antitumor immune response, but it is also possible that each of these complex products has a direct effect on the immune system that should be investigated. 

It has been widely discussed that the regulation of glucose consumption by tumor cells may become an interesting therapeutic strategy; however, some authors suggest that this may be detrimental to the antitumor immune response, particularly since T cells require glucose to exert their effector function [19,20]. In the present work, we observed that Anamu-SC reduces glucose consumption in tumor cells (Figure 2), but this apparently has no effect on the population of T cells producing IFNγ and TNFα, although it does reduce the frequency of T cells producing IL-2 (Figure 6). IFNγ production has previously been shown to be sensitive to glucose consumption [19], and also TNFα, but IL-2 production has not [21], so it could be inferred that Anamu-SC does not act on the T cells, but it is necessary to evaluate it.

In fact, the effect of these natural products on the immune response should be studied further, since although there are reports of anti-inflammatory [22], immunomodulatory [8], hypoglycemic, and antioxidant activity of Anamu [23,24], the evidence is not conclusive enough, and even less if these activities are related to immune response modulation. Indeed, agents like the metabolic regulator metformin redirect the metabolism of TAMs to lower OXPHOS and higher glycolysis in osteosarcoma, supporting anti-tumor effects [25]. Indeed, Anamu extracts have been reported to be antioxidants, but in our hands, when comparing the antioxidant activity of different batches of Anamu-SC extract versus the antioxidant activity of P2Et, Anamu-SC is 250 times less antioxidant than P2Et [9].

The interaction between tumor metabolism, immunometabolism, and tumor control are currently hot topics to be considered in the development of cancer therapies. Many metabolic modulators are in clinical studies in cancer [26], and plant extracts that have been reported as metabolic modulators are not far behind [27,28,29,30,31]. In summary, the immune system plays an important role in the control of tumors and in the response to chemotherapy or biological modulators [32]. Multiple interactions between natural products that may be beneficial to the patient have been documented [33]. They can inhibit the secretion of cytokines such as TGF-β, PGE2, or IL-10, promote the secretion of antitumor cytokines such as IFNγ, TNFα, or IL-1β, and reduce the population of Tregs, myeloid-derived suppressor cells (MDSCs), or M1 macrophages by increasing the effector T cell count [34,35,36]. The interaction between the various metabolites present in the standardized extracts of plants is studied through the analysis of the interaction of complex networks, which consider the multiple molecular weights that can contain these extracts [37]. The truth is that complex extracts can act with a high specificity that is conditioned by the variety of primary and secondary metabolites present in each mixture. Reverse pharmacology allows this analysis to be more accurate since it is based on the traditional use of the plant, from which the validation of its biological activity begins [38].

The efficacy of these herbal preparations is increasingly clear, but tumor heterogeneity conditions the response, just as it is for synthetic drugs, which speaks well of their specificity and of their future benefits but underlines the fact that each of these standardized extract drugs, currently called polymolecular drugs, must follow the rigorous development of synthetic drugs [38].

## 4. Materials and Methods

### 4.1. Plant Material

Leaves of *Petiveria alliacea* were collected in Quipile (Cundinamarca, Colombia) and identified by the Colombian National Herbarium (voucher number 333406). The Anamu-SC extract was obtained and chemically characterized as previously described [9]. Fresh pods of *Caesalpinia spinosa* were collected in Villa de Leyva, Boyacá, Colombia, and identified by the Colombian National Herbarium (voucher number COL 588448). The P2Et extract was produced from *Caesalpinia spinosa,* as previously reported [12]. The drug extraction ratio was 20–33.3:1, where the P2Et contains a high concentration of hydrolyzable tannins ranging from 70 to 95%. These tannins were present in the form of mono-, di-, and tri-O-galloyl quinic derivatives, with a calculated amount of 530% gallic acid and 2–7% methyl gallate and ethyl gallate. The P2Et extract was produced under Good Manufacturing Practices conditions. Furthermore, prior to conducting any experiments, the extract undergoes rigorous physicochemical and microbiological certification to guarantee consistency in its quality and consistency.

### 4.2. Ultra-High-Performance Liquid Chromatography (UPLC) Analysis

The UPLC analysis was carried out on an Acquity H Class UPLC Waters^®^ with a photodiode array detector eλ acquity, quaternary pump, degasser, and autosampler. Data were processed using Empower^®^ 3 software version 5.41.0. A BEH Shield C18 1.7 µm 2.1 mm × 100 mm Waters^®^ BEH Shield C18 column, maintained at 25 °C ± 1 with an elution gradient of acetonitrile (solvent A) and 0.1% (solvent B) as follows: 0% A (0–2 min), 50% A (9–10 min), 75% A (15–16 min), 90–100% A (21–26 min), 100% A (26–28 min), 50–0% A (30–40 min), with a run flow rate of 0.15 mL min^−1^ and an injection volume of 3 µL. For detection, the wavelength used was 274 nm, while the UV spectrum was monitored in the range of 200 to 400 nm.

### 4.3. In Vitro Cytotoxicity Assays

The Anamu-SC cytotoxic effect on tumor cells was evaluated using the methylthiazol tetrazolium (MTT) assay (Sigma-Aldrich, Saint Louis, MO, USA) as previously reported [7]. The IC_50_ value (50% inhibition of cell growth) was calculated using GraphPad Prism version 8.1.1 for Mac OS X statistics software (GraphPad Software, San Diego, CA, USA).

### 4.4. Proliferation Assay

B16-F10 and 4T1 murine mammary tumor cells were seeded in 12-well plates at a density of 26,000 cells/cm^2^ and treated with the IC_50_ or IC_50_/5 of Anamu-SC extract or P2Et (positive control). After 6, 12, and 24 h, cells were collected and counted with 0.4% trypan blue. Using the exponential growth method (Mathusian), the PDT was calculated through the formula Y = Y0*exp(k*x) of the GraphPad Prism version 8.1.1 for Mac OS X statistics software (GraphPad Software, San Diego, CA, USA).

### 4.5. ROS Measurement

ROS production was measured using 2′,7′ diclorodihidrofluoresceina diacetato (H_2_DCFDA) (Sigma Aldrich, Saint Louis, MO, USA) as previously reported [12]. Cells were treated for 6 and 12 h with the IC_50_ and IC_50_/5 of the Anamu-SC extract, or the positive and negative controls [12]. Cells were then acquired using a FACSAria II (BD) and analyzed with FlowJo v10.8.1 software (BD Life Sciences, Franklin Lakes, NJ, USA). Experiments were performed in triplicate on three independent experiments.

### 4.6. Glucose Uptake Assay

Cells were incubated for 6 and 12 h with the IC_50_ and IC_50_/5 of the Anamu-SC or P2Et extract (positive control), rotenone (positive control, 1 µM for 4T1 and 50 µM for B16-F10), DMSO, or ethanol (negative controls, 0.02%) [12]. After treatments, cells were incubated with 40 µM of 2-(N-(7-Nitrobenz-2-oxa-1,3-diazol-4-il)amino)-2-desoxiglucosa (2-NBDG) (Invitrogen Molecular Probes). Live versus dead cell discrimination was performed with labeling with PI (Sigma-Aldrich). Cells were acquired using a FACSAria II (BD) flow cytometer and analyzed with FlowJo v10.8.1 software (BD Life Sciences). Experiments were performed in triplicate on two independent experiments.

### 4.7. Mice and Tumor Cell Lines

Young (6 to 12 weeks old) female C57BL/6 (C57BL/6NCrl strain) and BALB/c (BALB/cAnNCrl strain) mice were provided from the Charles River Laboratories (Bar Harbor, ME, USA) and housed at the animal facilities of the Pontificia Universidad Javeriana (PUJ, Bogotá, Colombia) according to the protocols of the Ethics Committee of the Faculty of Sciences, PUJ, and the National and International Legislation for Live Animal Experimentation (Republic of Colombia, Resolution 08430, 1993). Each specific protocol was approved by the animal experimentation committee of PUJ (FUA-093-20). B16-F10 and 4T1 cell lines were cultured in RPMI-1640 (Eurobio, Toulouse, France) supplemented as previously reported [12].

### 4.8. Abs

For surface marker labeling, the cell suspension was incubated with anti-CD3 Pacific Blue, anti-CD8 PE Dazzle 594, anti-CD45 PE-Cy5, anti-Ly-6G PE-Cy7, anti-Ly-6C APC-Cy7, anti-PD-L1 PE, anti-PD-1 APC, CD11b Alexa Fluor 700, anti-CD4 Brilliant Violet 570, anti-CD44 PE-Cy7, CD25 APC (Biolegend, San Diego, CA, USA), and CD11c FITC (BD Biosciences, San José, CA, USA) fluorescent antibodies. For Treg cell assessment, intracellular labeling was performed with anti-FoxP3 Alexa Fluor 488 (BD Biosciences) and anti-CTLA-4 PE (Biolegend) antibodies. For intracellular cytokine evaluation, cells were stained with anti-IFNγ Alexa Fluor 700, TNFα PE-Cy7, IL-2 FITC (BD Biosciences), anti-perforin, and anti-granzyme B (Biolegend) antibodies (Appendix A).

### 4.9. Acute Toxicity Evaluation

Female BALB/cAnNCrl or C57BL/6NCrl mice were intraperitoneally (IP) inoculated with 2000 mg/kg of Anamu-SC extract. Lethal dose 50% (LD_50_) was calculated with Probit version 14 (Minitab Inc, State College, PA, USA). To ensure no toxicity, animals were treated with 90 mg/kg (B16-body weight) of Anamu-SC extract, which corresponds to at least 10 times lower doses than LD_50_. The dose of P2Et extract was used as previously reported [12].

### 4.10. In Vivo Tumor Development Experiments and Treatment

In vivo models were made as previously described [12]. Briefly, for the melanoma model, C57BL/6NCrl mice were inoculated subcutaneously (s.c.) with 1 × 10^5^ B16-F10 cells. For the breast cancer model, 1 × 10^4^ 4T1 cells were injected via s.c. into the right mammary fat pad of BALB/cAnNCrl mice [12]. To evaluate the effect of different treatments on tumor growth, 5 days after the inoculation of tumor cells, mice were treated with 75 mg/kg (B16-F10 melanoma model) or 18.7 mg/kg (4T1 mammary carcinoma model) body weight of P2Et extract, 90 mg/kg body weight of Anamu-SC, PBS (negative control), 1 mg/kg body weight of Dox (positive control), Dox plus P2Et, or Dox plus Anamu-SC. Mice were euthanized by CO_2_ inhalation, and then the spleen, tumor-draining lymph nodes, and tumor were removed and processed for subsequent assays.

### 4.11. Evaluation of Immune Populations by Flow Cytometry

Briefly, 1 × 10^6^ cells were stained with LIVE/DEAD Fixable Aqua. After washing with PBS 2% fetal bovine serum (FBS), the cells were stained with the surface antibodies at a final concentration of 1 μg/mL according to the designed multicolor panels. To identify regulatory T cells, cells were previously stained with anti-CD45, anti-CD3, anti-CD4, and anti-CD25 antibodies and fixed and permeabilized using the True Nuclear Transcription Factor Buffer Set (Biolegend) according to the manufacturer’s instructions. Then, cells were stained with anti-FoxP3 and anti-CTLA-4 antibodies. The cells were acquired by flow cytometry using the Cytek Aurora Cytometer (Cytek Biosciences, Fremont, CA, USA), and the results were analyzed using FlowJo v10.8.1 software (BD Life Sciences).

### 4.12. Evaluation of the Immune Response by Flow Cytometry

To evaluate the immune response, cells obtained from the spleen were cultured with phorbol 12-myristate 13-acetate (PMA) and ionomycin (P/I) or without a stimulus for 6 h as previously described [12]. The last 5 h of culture were performed in the presence of brefeldin A (1 μg/mL) (BD Pharmingen). Briefly, 1 × 10^6^ cells were stained with LIVE/DEAD Fixable, after washing with PBS containing 2% FBS, the cells were stained with anti-CD45, anti-CD3, anti-CD4, and anti-CD8 antibodies. Later, the cells were washed, fixed, and permeabilized for final staining with anti-IFNγ, anti-TNFα, anti-IL-2, anti-perforin, and anti-granzyme B. Finally, the cells were washed and resuspended in PBS. Cells were acquired through flow cytometry using the Cytek Aurora Cytometer (Cytek Biosciences), and the data were subsequently analyzed using FlowJo v10.8.1 software (BD Life Sciences). Multifunctional analyses were performed using a Boolean gating strategy. The data are presented using Pestle v2.0 and SPICE v6.1 software (the National Institutes of Health, Bethesda, MD, USA) [39].

### 4.13. Cytokine Assay

Serum was obtained from blood drawn by cardiac puncture, and cytokine assessment was performed using a Cytometric Bead Array (CBA) mouse Th1, Th2, and Th17 cytokine kit (BD Biosciences) according to the manufacturer’s instructions. Events were acquired using a FACSAria IIU flow cytometer (BD Immunocytometry Systems), and the data were subsequently analyzed using FCAP array software version 3.0 (BD Biosciences).

### 4.14. Statistical Analysis

Statistical analysis between two groups was calculated using the Mann–Whitney U test, while differences among groups were calculated using Kruskal–Wallis and Dunn’s posttest for multiple comparisons. GraphPad Prism version 8.1.1 for Mac OS X statistics software (GraphPad Software) was used.

## Figures and Tables

**Figure 1 ijms-24-16698-f001:**
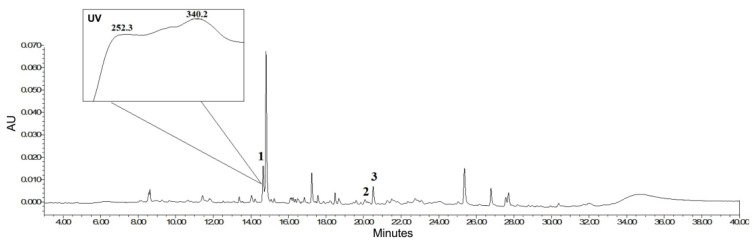
Chromatographic analysis of Anamu-SC extract at 254 nm and UV spectra. UPLC-PDA analysis was performed with an Acquity UPLC H-class (Waters, Milford, MA, USA) and Waters C 18 column (100 × 2.1 mm, 1.7 µm). Peak identification: (1) myricetin; (2) dibenzyl disulfide; (3) dibenzyl trisulfide. AU: absorbance units.

**Figure 2 ijms-24-16698-f002:**
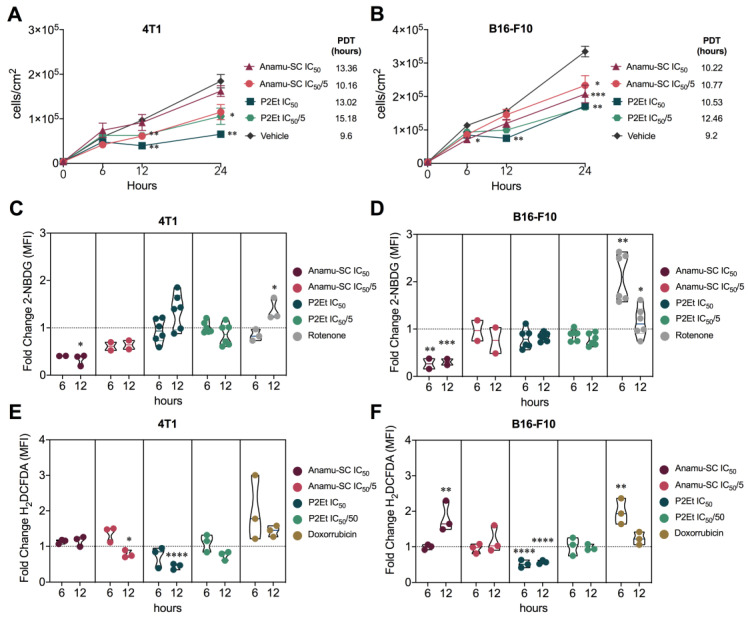
In vitro activity of Anamu-SC extract. Cell count per cm^2^ of (**A**) 4T1 and (**B**) B16-F10 cells after treatments for 0, 6, 12, and 24 h. Population doubling times (PDT) are shown in each legend. Fold change of 2-NBDG MFI after treatments with IC_50_ and IC_50_/5 of Anamu-SC extract, IC_50_ and IC_50_/5 of P2Et extract, or rotenone (positive control) for 6 and 12 h in 4T1 (**C**) and B16-F10 (**D**) cells. Fold change of 2NBDG MFI after the treatments with IC_50_ and IC_50_/5 of Anamu-SC, IC_50_ and IC_50_/5 of P2Et extract, or IC_50_ of doxorubicin (positive control) for 6 and 12 h in 4T1 (**E**) and B16-F10 (**F**) cells. In all cases, fold change was determined using the MFI of each treatment relative to its vehicle (ethanol or DMSO). H_2_DCFDA: 2-(N-(7-Nitrobenz-2-oxa-1,3-diazol-4-il)amino)-2-desoxiglucosa; MFI: mean fluorescence intensity; 2NBDG: 2′,7′ diclorodihidrofluoresceina diacetate. * *p* < 0.05; ** *p* < 0.01; *** *p* < 0.001; **** *p* < 0.0001.

**Figure 3 ijms-24-16698-f003:**
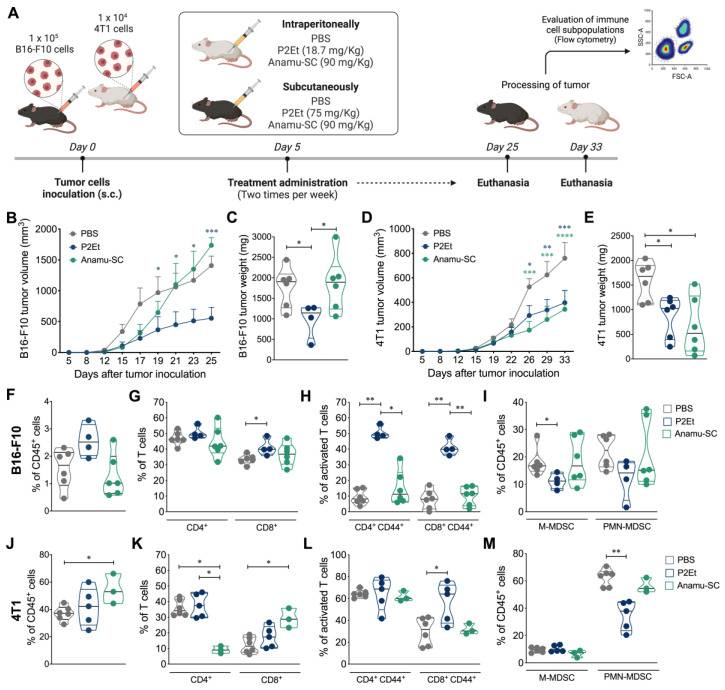
In vivo effect of Anamu-SC extract. (**A**). Experimental design to evaluate the effect of Anamu-SC extract in 4T1 and B16-F10 tumor-bearing mice. The tumor was established by subcutaneous injection of 4T1 cells for the mammary carcinoma model or B16-F10 for the melanoma model. Treatments were administered twice weekly from day 5 after tumor cell injection until the end of the experiment. (**B**). Tumor volume of B16-F10 tumor-bearing mice treated with each treatment. (**C**). B16-F10 tumor weight. (**D**). Tumor volume of 4T1 tumor-bearing mice treated with each treatment. (**E**). 4T1 tumor weight. (**F**). Frequency of B16-F10 intratumor CD45^+^ cells in mice treated with PBS (negative control), P2Et (positive control), or Anamu-SC extract. Frequency of CD4+ and CD8+ T cells (**G**), activated (CD44+) CD4+ and CD8+ T cells (**H**), and M-MDSC-LC and PMN-MDSC-LC (**I**), infiltrating B16-F10 tumor. (**J**). Frequency of 4T1 intratumor CD45^+^ cells in each group of mice. Frequency of CD4+ and CD8+ T cells (**K**), activated (CD44+) CD4+ and CD8+ T cells (**L**), and M-MDSC-LC and PMN-MDSC-LC (**M**), infiltrating the 4T1 tumor. Data are presented by violin plots showing all points with their corresponding median. The *p* values were calculated using the Mann–Whitney *U* test or Kruskal–Wallis and Dunn’s posttest for multiple comparisons. s.c.: subcutaneously; M-MDSC-LC: monocytic myeloid-derived suppressor-like cells; PMN-MDSC-LC: polymorphonuclear myeloid-derived suppressor-like cells. * *p* < 0.05, ** *p* < 0.01, *** *p* < 0.001, **** *p* < 0.0001.

**Figure 4 ijms-24-16698-f004:**
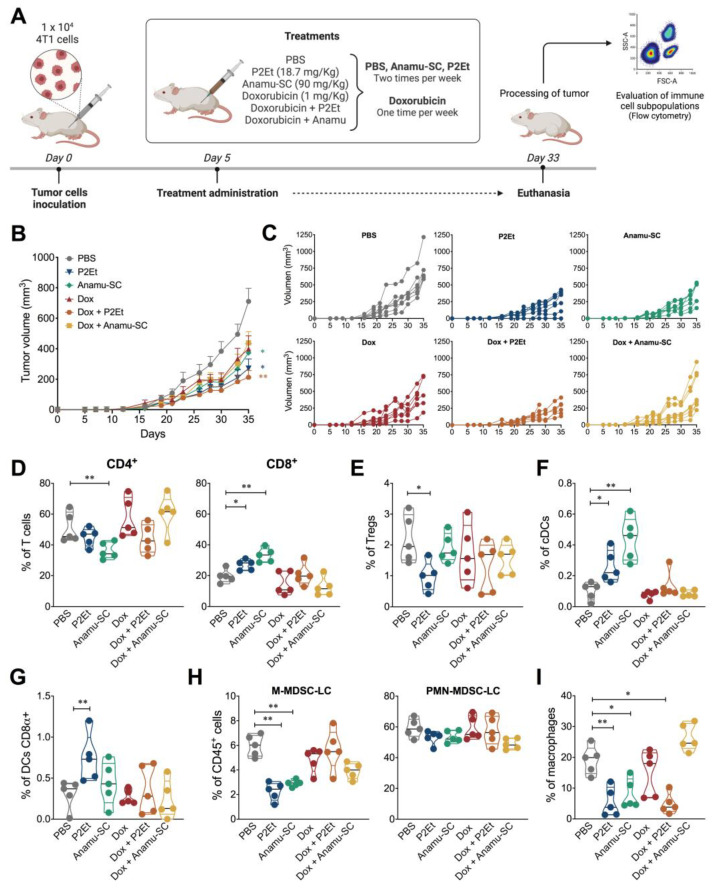
In vivo effect of doxorubicin co-treatments in the murine 4T1 murine mammary carcinoma model. (**A**). Experimental design to evaluate the effect of Dox co-treatments in the 4T1 tumor-bearing mice. The tumor was established by subcutaneous injection of 4T1 cells, and treatments were administered from day 5 after tumor cell injection until the end of the experiment. (**B**). Tumor volume of 4T1 tumor-bearing mice treated with each treatment. (**C**). Tumor volume growth curves for individual mice in each treatment group. (**D**). Frequency of CD4+ and CD8+ T cells (**D**), Tregs (**E**), conventional DCs (**F**), CD8α DCs (**G**), M-MDSC-LC and PMN-MDSC-LC (**H**), and macrophages (**I**) infiltrating the 4T1 tumor. Data are presented by violin plots showing all points with their corresponding median. The *p* values were calculated using the Mann–Whitney *U* test or Kruskal–Wallis and Dunn’s posttest for multiple comparisons. cDCs: convectional dendritic cells; DCs: dendritic cells; Dox: doxorubicin; M-MDSC-LC: monocytic myeloid-derived suppressor-like cells; PMN-MDSC-LC: polymorphonuclear myeloid-derived suppressor-like cells; Tregs: regulatory T cells. * *p* < 0.05, ** *p* < 0.01.

**Figure 5 ijms-24-16698-f005:**
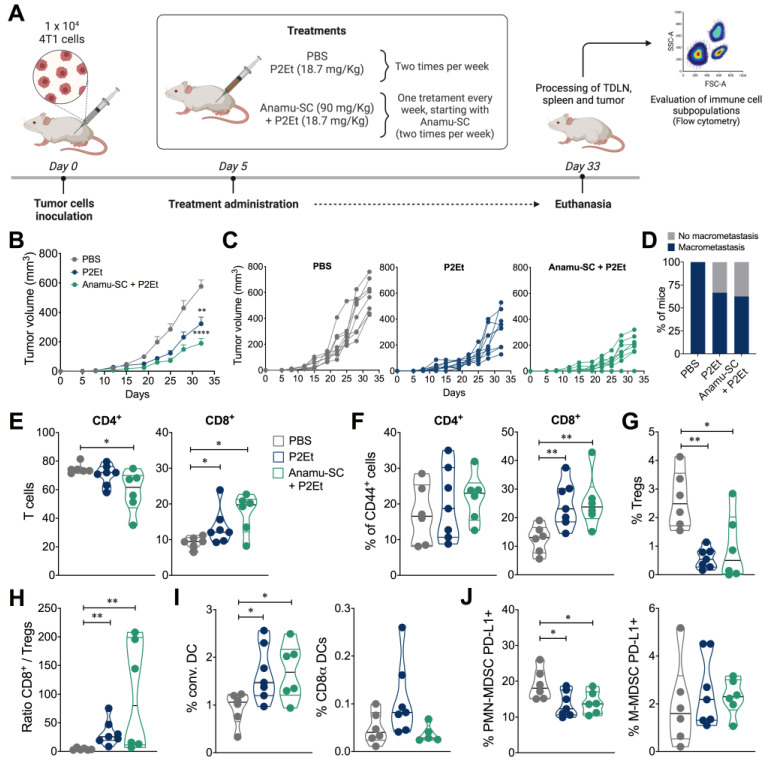
In vivo effect of co-treatments of Anamu-SC and P2Et in the murine 4T1 murine mammary carcinoma model. (**A**). Experimental design to evaluate the effect of co-treatment with Anamu-SC and P2Et in the 4T1 tumor-bearing mice. The tumor was established by subcutaneous injection of 4T1 cells, and treatments were administered two times per week from day 5 post-inoculation until the end of the experiment. (**B**). Tumor volume of 4T1 tumor-bearing mice. (**C**). Tumor volume growth curves for individual mice in each treatment group. (**D**). Bars show the percentage of mice that developed macrometastasis. Frequency of CD4+ and CD8+ T cells (**E**), activated CD4+ and CD8+ T cells (**F**), Tregs (**G**), ratio CD8/Tregs (**H**), conventional and CD8α DCs (**I**), and M-MDSC-LC and PMN-MDSC-LC (**J**) infiltrating 4T1 tumor. Data are presented by violin plots showing all points with their corresponding median. The *p* values were calculated using the Mann–Whitney *U* test or Kruskal–Wallis and Dunn’s posttest for multiple comparisons. conv. DCs: convectional dendritic cells; DCs: dendritic cells; M-MDSC-LC: monocytic myeloid-derived suppressor like cells; PMN-MDSC-LC: polymorphonuclear myeloid-derived suppressor like cells; Tregs: regulatory T cells. * *p* < 0.05, ** *p* < 0.01, **** *p* < 0.001.

**Figure 6 ijms-24-16698-f006:**
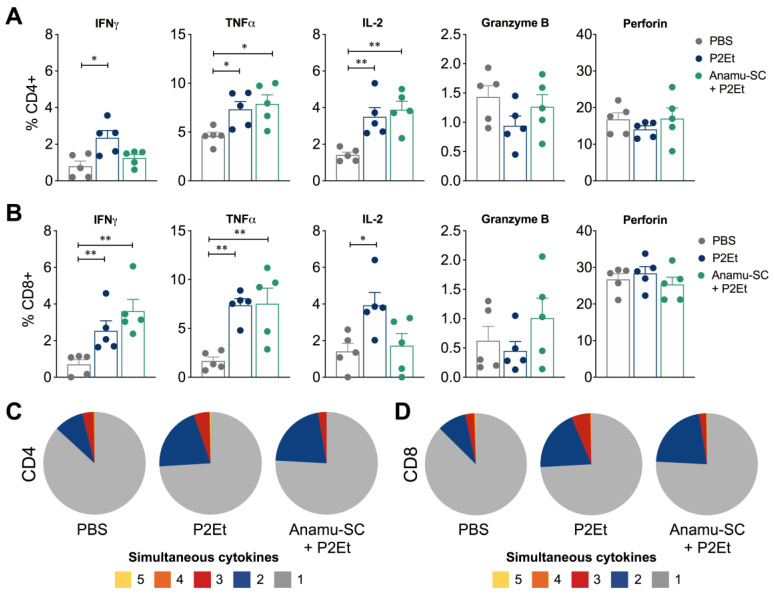
Functional activity of the T cells. Frequency of CD4+ (**A**) or CD8+ (**B**) T cells from the spleen producing IFNγ, TNFα, IL-2, granzyme B, or perforin following stimulation with phorbol 12-myristate 13-acetate (PMA)/ionomycin (P/I). Polyfunctional activity of CD4+ (**C**) or CD8+ (**D**) T cells, following stimulation with P/I, from each group of treatment. As shown in the pie charts, the functional profiles are grouped and color-coded according to the number of functions. Data are presented by violin plots showing all points with their corresponding median. The *p* values were calculated using the Mann–Whitney *U* test. PMA: phorbol 12-myristate 13-acetate. * *p* < 0.05, ** *p* < 0.01.

## Data Availability

The data presented in this study are available in this article (and Appendix A).

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
