# Peer review of "Natural Products Induce Different Anti-Tumor Immune Responses in Murine Models of 4T1 Mammary Carcinoma and B16-F10 Melanoma"

_ijms, 2023, doi:10.3390/ijms242316698_

Round 1

Reviewer 1 Report

Comments and Suggestions for Authors

The  work is well designed and presented. However, there are several inaccuracies:

 The model of the work is often repeated in other articles of the group. Proofread the materials and methods thoroughly.

 Methodology described as meaningless, since it was not executed- ROS, annexin V (collage to other works)

 Text with repetition in other published manuscripts

The acute toxicity found, even if divided by 4, can give toxicity in the animal. Obtaining the NOAEL would be the most correct. If they did not do so, they should ascertain the toxicity with the determination of organ biomarkers. Thus, the result may be influenced by some toxicity.

Cell viability should be better explained. The values found are not understood.

 Abbreviations should be considered, since the materials and methods are at the end of the manuscript (Expl: Fig. 2)

 The discussion should be followed with reference of the Figs, for context!

Review the bibliography. Ref. 12 is incomplete....

 In the certainty of the absence of toxicity on the part of the extracts, the conclusions would be more precise. The authors should take into account that total extracts have multiple bioactive compounds, which may bind to different cell receptors, exhibiting different activities.

 The  work is well designed and presented. However, there are several inaccuracies:

 The model of the work is often repeated in other articles of the group. Proofread the materials and methods thoroughly.

 Methodology described as meaningless, since it was not executed- ROS, annexin V (collage to other works)

 Text with repetition in other published manuscripts

The acute toxicity found, even if divided by 4, can give toxicity in the animal. Obtaining the NOAEL would be the most correct. If they did not do so, they should ascertain the toxicity with the determination of organ biomarkers. Thus, the result may be influenced by some toxicity.

Cell viability should be better explained. The values found are not understood.

 Abbreviations should be considered, since the materials and methods are at the end of the manuscript (Expl: Fig. 2)

 The discussion should be followed with reference of the Figs, for context!

Review the bibliography. Ref. 12 is incomplete....

 In the certainty of the absence of toxicity on the part of the extracts, the conclusions would be more precise. The authors should take into account that total extracts have multiple bioactive compounds, which may bind to different cell receptors, exhibiting different activities.

The  work is well designed and presented. However, there are several inaccuracies:

 The model of the work is often repeated in other articles of the group. Proofread the materials and methods thoroughly.

 Methodology described as meaningless, since it was not executed- ROS, annexin V (collage to other works)

 Text with repetition in other published manuscripts

The acute toxicity found, even if divided by 4, can give toxicity in the animal. Obtaining the NOAEL would be the most correct. If they did not do so, they should ascertain the toxicity with the determination of organ biomarkers. Thus, the result may be influenced by some toxicity.

Cell viability should be better explained. The values found are not understood.

 Abbreviations should be considered, since the materials and methods are at the end of the manuscript (Expl: Fig. 2)

 The discussion should be followed with reference of the Figs, for context!

Review the bibliography. Ref. 12 is incomplete....

 In the certainty of the absence of toxicity on the part of the extracts, the conclusions would be more precise. The authors should take into account that total extracts have multiple bioactive compounds, which may bind to different cell receptors, exhibiting different activities.

Author Response

We thank you for your thoughtful and constructive comments. Please find attached our response in which we have addressed the comments point by point:

  1. The model of the work is often repeated in other articles of the group. Proofread the materials and methods thoroughly.

The model described in methodology was reviewed and a previous article by the group with this methodology was cited. Additionally, the text was modified.

  1. Methodology described as meaningless, since it was not executed- ROS, annexin V (collage to other works)

According to the evaluator's assessment, the annexin methodology was set by mistake, so it was eliminated, however, we kept the ROS methodology because it is part of the experiments in the paper.

  1. Text with repetition in other published manuscripts

Parts of the methodology were modified to avoid repetition with other publications

  1. The acute toxicity found, even if divided by 4, can give toxicity in the animal. Obtaining the NOAEL would be the most correct. If they did not do so, they should ascertain the toxicity with the determination of organ biomarkers. Thus, the result may be influenced by some toxicity.

Experimentally we have not considered the calculation of the NOAEL due to the 3R principles. However, we appreciate the evaluator's observation and will take his recommendation into account for future studies, knowing that it is the best option.

Additionally, we corrected the text because the concentration used was at least 10 times lower than the lethal dose 50, as previously reported in the following reference. However, it is important to mention that with the doses we have used we have not found toxicity in any of the organs examined during the postmortem analysis.

Hernández, J. F., Urueña, C. P., Sandoval, T. A., Cifuentes, M. C., Formentini, L., Cuezva, J. M., & Fiorentino, S. (2017). A cytotoxic Petiveria alliacea dry extract induces ATP depletion and decreases β-F1-ATPase expression in breast cancer cells and promotes survival in tumor-bearing mice. Revista Brasileira de Farmacognosia, 27, 306-314.

  1. Cell viability should be better explained. The values found are not understood.

Supplementary Figure 1 includes the dose-response viability curves with which the IC50 of the extracts for each cell line is determined. Additionally, a sentence was included in the results section better explaining the meaning of the concentration values.

  1. Abbreviations should be considered, since the materials and methods are at the end of the manuscript (Expl: Fig. 2)

According to the reviewer's suggestion, the acronyms are described in Figure 2. Additionally, all acronyms were reviewed and adjusted.

  1. The discussion should be followed with reference of the Figs, for context!

According to the reviewer's suggestion, figures were included in the discussion according to the result discussed.

  1. Review the bibliography. Ref. 12 is incomplete....

The bibliography was reviewed and information on some references was completed.

  1. In the certainty of the absence of toxicity on the part of the extracts, the conclusions would be more precise. The authors should take into account that total extracts have multiple bioactive compounds, which may bind to different cell receptors, exhibiting different activities.

We appreciate the comment, and in fact the complexity of phytomedicines means that their study must consider several factors such as molecular diversity, traditional use and artificial intelligence tools that are currently being developed to do network pharmacology and reverse pharmacology. A reflection on this topic was added to the discussion.

Reviewer 2 Report

Comments and Suggestions for Authors

The research presented in the article "Natural products induce different anti-tumor immune responses in the 4T1 breast cancer and B16-F10 melanoma models" aims to elucidate the differences in responses to Anamu-SC and P2Et treatments in two distinct tumor models. It is demonstrated that the Anamu-SC extract reduces tumor growth in the 4T1 breast cancer model but not in the B16-F10 melanoma model, unlike the standardized P2Et extract. Both extracts reduced the levels of IL-10 in the B16-F10 model, but only P2Et increased the levels of TNFα and IFNγ. Analysis of intratumoral immune infiltrates revealed that Anamu-SC decreased the frequency of CD4+ T cells more than P2Et, but it markedly increased the frequency of CD8+ T cells. The findings shed light on the effectiveness of these herbal preparations and highlight the specificity influenced by tumor heterogeneity and the distinct chemical complexity of each preparation.

While these results contribute to our understanding of specificity and its potential benefits, they also underscore the necessity for a rigorous developmental pathway for each of these standardized extracts, referred to as polymolecular drugs, to elucidate their biological activity. However, the article requires minor corrections and clarification of inaccuracies.

1. Have the authors checked the effects of the mentioned extracts on other melanoma and breast cancer lines? Are the presented results specific to scell lines or to a type of cancer?

2. Has the effect of extracts on cell viability been previously tested on normal cells? For example on normal mouse fibroblasts?

3. In figure 2E and 2F data for 24h is not shown on the diagrams.

4. The introduction should include a short description of the composition of the P2Et extract.

Author Response

We thank you for your thoughtful and constructive comments. Please find attached our response in which we have addressed the comments point by point:

  1. Have the authors checked the effects of the mentioned extracts on other melanoma and breast cancer lines? Are the presented results specific to scell lines or to a type of cancer?

We have tested the effect of the extracts on other cell lines, including human and murine breast cancer, melanoma and leukemia cell lines, finding an important effect on them. We include some of the related publications:

- Urueña, C., Sandoval, T. A., Lasso, P., Tawil, M., Barreto, A., Torregrosa, L., & Fiorentino, S. (2020). Evaluation of chemotherapy and P2Et extract combination in ex-vivo derived tumor mammospheres from breast cancer patients. Scientific Reports, 10(1), 19639.

- Gomez-Cadena, A., Urueña, C., Prieto, K. et al. Immune-system-dependent anti-tumor activity of a plant-derived polyphenol rich fraction in a melanoma mouse model. Cell Death Dis 7, e2243 (2016). https://doi.org/10.1038/cddis.2016.134

- Sandoval, T. A., Urueña, C. P., Llano, M., Gómez-Cadena, A., Hernández, J. F., Sequeda, L. G., ... & Fiorentino, S. (2016). Standardized extract from Caesalpinia spinosa is cytotoxic over cancer stem cells and enhance anticancer activity of doxorubicin. The American journal of Chinese medicine, 44(08), 1693-1717.

- Hernández, J. F., Urueña, C. P., Sandoval, T. A., Cifuentes, M. C., Formentini, L., Cuezva, J. M., & Fiorentino, S. (2017). A cytotoxic Petiveria alliacea dry extract induces ATP depletion and decreases β-F1-ATPase expression in breast cancer cells and promotes survival in tumor-bearing mice. Revista Brasileira de Farmacognosia, 27, 306-314.

- Murillo, N., Lasso, P., Urueña, C., Pardo-Rodriguez, D., Ballesteros-Ramírez, R., Betancourt, G., ... & Fiorentino, S. (2023). Petiveria alliacea Reduces Tumor Burden and Metastasis and Regulates the Peripheral Immune Response in a Murine Myeloid Leukemia Model. International Journal of Molecular Sciences, 24(16), 12972.

  1. Has the effect of extracts on cell viability been previously tested on normal cells? For example on normal mouse fibroblasts?

We have also tested the effect of the extracts on cell viability in non-tumor cells, such as 3T3 fibroblasts, and we have not found a significant effect on their viability, finding high selectivity index values for tumor lines. Some of the articles where these results are reported are:

- Hernández, J. F., Urueña, C. P., Sandoval, T. A., Cifuentes, M. C., Formentini, L., Cuezva, J. M., & Fiorentino, S. (2017). A cytotoxic Petiveria alliacea dry extract induces ATP depletion and decreases β-F1-ATPase expression in breast cancer cells and promotes survival in tumor-bearing mice. Revista Brasileira de Farmacognosia, 27, 306-314.

- Murillo, N., Lasso, P., Urueña, C., Pardo-Rodriguez, D., Ballesteros-Ramírez, R., Betancourt, G., ... & Fiorentino, S. (2023). Petiveria alliacea Reduces Tumor Burden and Metastasis and Regulates the Peripheral Immune Response in a Murine Myeloid Leukemia Model. International Journal of Molecular Sciences, 24(16), 12972.

  1. In figure 2E and 2F data for 24h is not shown on the diagrams.

The error in the figure legend and in the methodology was corrected.

  1. The introduction should include a short description of the composition of the P2Et extract.

Following the evaluator's recommendation, we included a small paragraph with the description of the extract.